# Motor planning brings human primary somatosensory cortex into action-specific preparatory states

**Giacomo Ariani**[1,2]*, **J Andrew Pruszynski**[1,3,4,5], **Jörn Diedrichsen**[1,2,6]

[1]The Brain and Mind Institute, Western University, London, Canada; [2]Department of Computer Science, Western University, London, Canada; [3]Department of Physiology and Pharmacology, Western University, London, Canada; [4]Department of Psychology, Western University, London, Canada; [5]Robarts Research Institute, Western University, London, Canada; [6]Department of Statistical and Actuarial Sciences, Western University, London, Canada

**Abstract** Motor planning plays a critical role in producing fast and accurate movement. Yet, the neural processes that occur in human primary motor and somatosensory cortex during planning, and how they relate to those during movement execution, remain poorly understood. Here, we used 7T functional magnetic resonance imaging and a delayed movement paradigm to study single finger movement planning and execution. The inclusion of no-go trials and variable delays allowed us to separate what are typically overlapping planning and execution brain responses. Although our univariate results show widespread deactivation during finger planning, multivariate pattern analysis revealed finger-specific activity patterns in contralateral primary somatosensory cortex (S1), which predicted the planned finger action. Surprisingly, these activity patterns were as informative as those found in contralateral primary motor cortex (M1). Control analyses ruled out the possibility that the detected information was an artifact of subthreshold movements during the preparatory delay. Furthermore, we observed that finger-specific activity patterns during planning were highly correlated to those during execution. These findings reveal that motor planning activates the specific S1 and M1 circuits that are engaged during the execution of a finger press, while activity in both regions is overall suppressed. We propose that preparatory states in S1 may improve movement control through changes in sensory processing or via direct influence of spinal motor neurons.

*For correspondence: giacomo.ariani@gmail.com

## Editor's evaluation

In this elegant and rigorous study, the authors investigated the neural correlates of planning and execution of single finger presses in a 7T fMRI study focusing on primary somatosensory (S1) and motor (M1) cortices. BOLD patterns of activation/deactivation and finger-specific pattern discriminability indicate that M1 and S1 are involved not only during execution, but also during planning of single finger presses. These important results clearly establish that the role of primary somatosensory cortex goes beyond pure processing of tactile information and will be of great interest for researchers in the field of motor control and of systems neuroscience.

## Introduction

Animals are capable of generating a wide variety of dexterous behaviors accurately and effortlessly on a daily basis. This remarkable ability relies on the motor system reaching the appropriate preparatory state before each movement is initiated.

At the level of behavior, the process of motor programming, or planning, has long been shown to be beneficial to performance (*Keele, 1968*; *Keele et al., 1976*; *Rosenbaum, 1980*), leading to faster reaction times (*Klapp and Erwin, 1976*; *Klapp, 1995*; *Haith et al., 2016*) and more accurate response selection (*Ghez et al., 1997*; *Wong and Haith, 2017*; *Ariani and Diedrichsen, 2019*; *Hardwick et al., 2019*). The behavioral study of motor planning led to neurophysiological investigations showing the presence of preparatory signals in the patterns of neuronal firing in the dorsal premotor cortex (PMd, *Cisek and Kalaska, 2004*; *Cisek and Kalaska, 2010*; *Hoshi and Tanji, 2006*), the supplementary motor area (*Hoshi and Tanji, 2004*), and the posterior parietal cortex (*Cui and Andersen, 2007*; *Cui and Andersen, 2011*; *Andersen and Cui, 2009*). Building on this work, human neuroimaging studies have shown that activity in parietofrontal brain regions during planning of prehension movements can be used to decode several movement properties such as grip type (*Gallivan et al., 2011b*; *Ariani et al., 2015*), action order (*Gallivan et al., 2016*), and effector (*Gallivan et al., 2011a*, *Gallivan et al., 2013*; *Leoné et al., 2014*; *Turella et al., 2016*).

At the level of neural population dynamics (*Vyas et al., 2020*), motor planning can be understood as bringing the neuronal state toward an ideal preparatory point. Once this state is reached and the execution is triggered, the intrinsic dynamics of the system then let the movement unfold (*Churchland et al., 2010*; *Shenoy et al., 2013*). Preparatory neural processes have not only been observed in premotor and parietal areas, but also in primary motor cortex (M1, *Tanji and Evarts, 1976*; *Crammond and Kalaska, 2000*; *Ariani et al., 2018*). In contrast, the degree to which primary somatosensory cortex (S1) receives information about the planned movement before movement onset is less clear.

S1 is often considered to be mostly concerned with processing incoming sensory information from tactile and proprioceptive receptors arising after movement onset. Consistent with this notion, previous functional magnetic resonance imaging (fMRI studies have not detected the presence of planning-related information in this area *Gallivan et al., 2011a*; *Gallivan et al., 2011b*, *Gallivan et al., 2016*; *Gallivan et al., 2013*; although see *Gale et al., 2021*). However, challenging this notion, in the past years research has shown that S1 can be somatotopically activated even in the absence of tactile inputs, for instance during touch observation (*Kuehn et al., 2014*), attempted movements without afferent tactile inputs (*Kikkert et al., 2021*), and through attentional shifts (*Puckett et al., 2017*). Moreover, a recent human electrocorticography (ECoG) study suggested a possible role for S1 in cognitive-motor imagery (*Jafari et al., 2020*). The authors recorded neural activity from S1 while a tetraplegic participant imagined reaching movements and found that S1 neurons encoded movement direction during motor imagery in the absence of actual sensations. Another recent ECoG study in nonhuman primates (*Umeda et al., 2019*) showed grasp-specific information in the signals from S1 well before movement initiation, and only slightly later than in M1.

However, it remains unknown whether S1 plays a role during motor planning in human participants with an intact sensory system. Furthermore, we currently do not know how the signals during action preparation relate to those during execution, a fact that could provide important insight into the role these signals may play.

Here, we designed a high-field (7T) fMRI experiment to study what brain regions underlie the planning of individual finger presses and how these brain representations relate to those during execution. We used variable delays between an instructing cue and a go signal, as well randomly interspersed no-go trials, to temporally separate the evoked responses to movement planning and execution. Using advanced multivariate pattern analyses we were able to examine the relationship between the fMRI patterns related to planned and executed finger actions.

## Results

### Deactivation in sensorimotor regions during planning of finger actions

We instructed 22 participants to plan and execute repeated keypresses with individual fingers of their right hand on a keyboard device while being scanned with 7T fMRI. The key to be pressed corresponded to one of three fingers and was cued during the preparation phase by numbers (1 = thumb, 3 = middle, 5 = little, e.g., *Figure 1A*) presented on a computer screen that was visible to the participants lying in the scanner through an angled mirror. After a variable delay (4–8 s), participants received a color cue indicating whether to press the planned finger (go trials), or whether to withhold

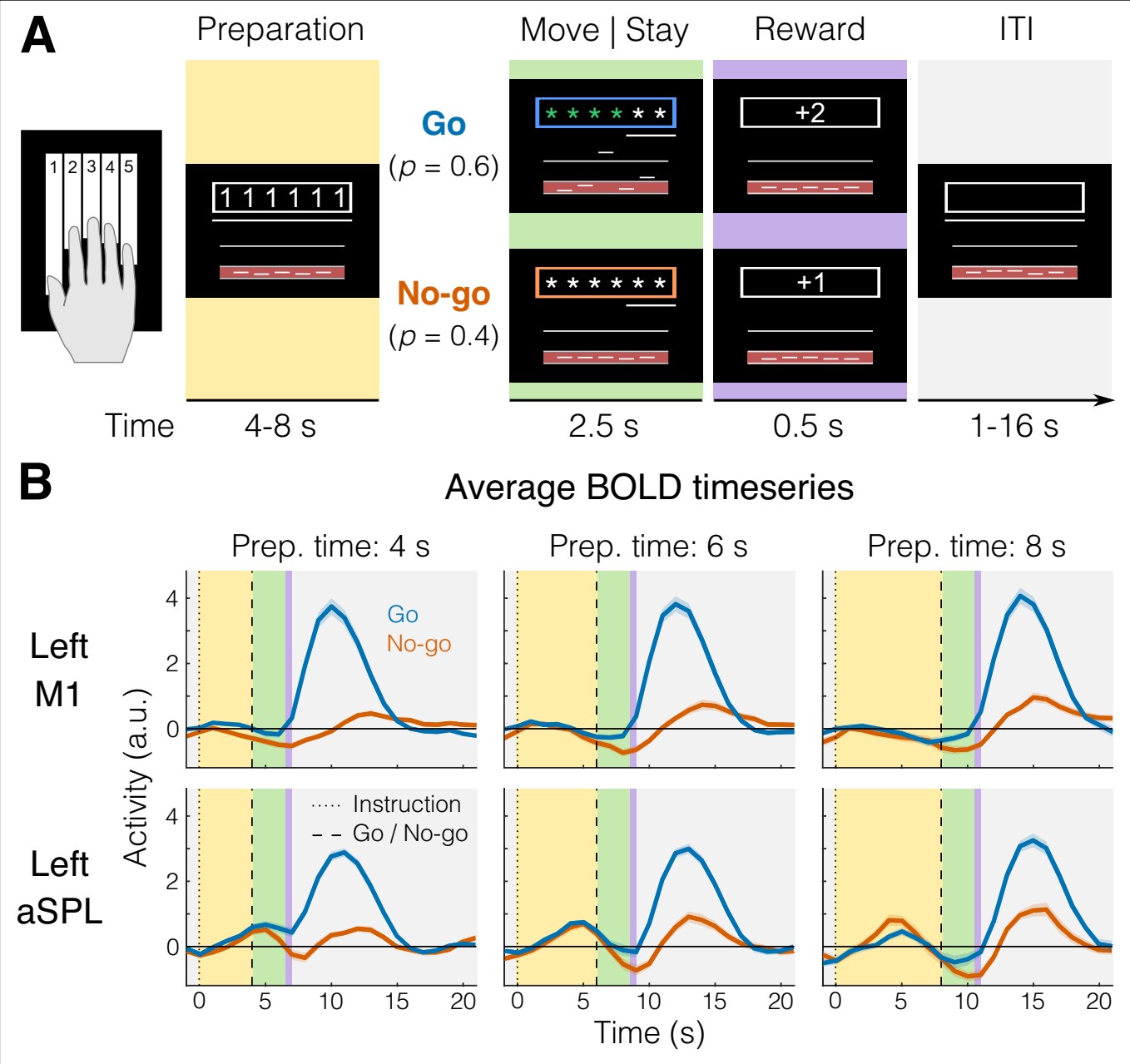

**Figure 1.** fMRI task and blood-oxygen-level-dependent (BOLD) responses.
(**A**) Example trial with timing information. Background colors indicate different experimental phases (yellow = preparation; green = move [go] or stay [no-go]; purple = reward; gray = intertrial interval, ITI). (**B**) Group-averaged BOLD response (*N* = 22) for go (blue) and no-go (orange) trials in a region that shows no planning-evoked activity (left M1, top), and one that shows some planning-evoked activity (left anterior superior parietal lobule [aSPL], bottom). Shaded areas indicate standard error of the mean (SEM). Background colors correspond to trial phases as in A.

the response (no-go trials). Upon the go cue, participants had to initiate the correct response as fast as possible and make six presses of the designated finger, before receiving accuracy points for reward (see Materials and methods).

To control for involuntary overt movements during the preparation phase, we required participants to maintain a steady force on all the keys during the delay, which was closely monitored online. To ensure that planning results would not be biased by the subsequent execution, we restricted all our analyses of the preparation phase to no-go trials only (see Materials and methods). First, we asked which brain regions showed an evoked response during the planning of finger presses (e.g., *Figure 1B*).

We focused our analysis on the lateral aspect of the contralateral (left) hemisphere (purple and white areas of *Figure 2* inset), which included the primary motor and somatosensory cortex, as well as the premotor cortex and anterior parietal cortical regions. To examine brain activation during finger planning, we performed a univariate contrast of the preparation phase (across the three fingers) vs the resting baseline (*Figure 2A*). Overall, the instruction stimulus evoked little to no activation in our regions of interest (ROIs, see Materials and methods). In fact, compared to resting baseline, we observed significant deactivation (*Figure 2E*) in the primary motor cortex (M1, $t_{21}$ = −6.939, p = 7.446e−07) and in the primary somatosensory cortex (S1, $t_{21}$ = −5.508, p = 1.823e−05). Significant deactivation was also observed in the PMd ($t_{21}$ = −2.929, p = 0.008). While anterior regions in the superior parietal lobule (aSPL) showed some signs of activation (*Figures 1B and 2A*), these did not reach statistical significance when tested at the ROI level ($t_{21}$ = 1.881, p = 0.074).

A wider whole-brain search (*Figure 2—figure supplement 2*) did not provide evidence for planning-related activation in other secondary motor areas. This lack of planning-related activation in high-order areas in planning is likely explained by the low task difficulty (i.e., little planning demands). Participants were only asked to plan repeated movements of a single finger, resulting in little amounts of overall planning activation. In contrast, execution strongly activated both primary and high-order sensorimotor regions (*Figure 2C*), with activation being significant in all tested ROIs (*Figure 2E*, all $t_{21}$ > 14.824, all p < 1.351e−12).

## Planning induces informative patterns in primary somatosensory and motor cortex

Although we found little univariate planning-related activation, preparatory processes need not increase the overall activation in a region. Rather, the region could converge to a specific preparatory neural state (*Churchland et al., 2010*), while activity increments and decrements within the region (i.e., at a finer spatial scale) average each other out. In this case, information about planned movements would be present in the multivoxel activity patterns in that region.

To test this idea, we calculated the cross-validated Mahalanobis dissimilarity, or crossnobis distance (see Materials and methods), between activity patterns. First, the activation patterns (beta weights) for the planning phase of no-go trials where prewhitened using the voxel-by-voxel covariance matrix. The distance was then calculated by comparing activity patterns across partitions (imaging runs), such that the value of the dissimilarity is on average zero if the two conditions only differ by measurement noise. Thus, systematically positive values of this dissimilarity measure indicate that the patterns reliably differentiate between the different planned actions (*Walther et al., 2016*; *Arbuckle et al., 2020*). Indeed, a surface-based searchlight approach (*Oosterhof et al., 2012*) revealed reliably positive crossnobis distance between the activity patterns related to planning of individual finger presses (*Figure 2B*), which the ROI analysis confirmed to be significantly greater than zero in both M1 ($t_{21}$ = 2.343, p = 0.029) and S1 ($t_{21}$ = 3.137, p = 0.005, *Figure 2F*).

The distribution on the flat surface map of these distance values during planning (*Figure 2B*) appeared to be highly similar to the distribution of distances during execution (*Figure 2D*). To quantify this topographic similarity, we computed the ratio between distances in different Brodmann area (BA) subdivisions of our ROIs (see Materials and methods), reasoning that a mismatch in location would result in large differences in ratio values. However, the ratio between planning and execution distances was roughly stable across the different subregions of sensorimotor cortex (BA 4a: 0.23, BA 4b: 0.16, BA 3a: 0.24, BA 3b: 0.19, BA 1: 0.22, BA 2: 0.31). In other words, the average distance between finger-specific activity patterns during planning was between 16% and 31% of the average pattern distance during execution. Thus, we not only show the existence of planning-related activity in S1, but also that S1 activity patterns are at least as informative as M1 activity patterns.

Visual inspection suggested that the informative patterns during planning may be concentrated more dorsally in M1 and S1 relative to execution. To test for the possibility that the location on the flat surface map of the peaks of the crossnobis distance for M1 and S1 was statistically different across subjects between planning and execution, we used a Hotelling $T^2$ test that allowed us to compare the difference between two multivariate means of different distributions (i.e., the distributions of x–y coordinates for the peaks of planning and execution). This test revealed no systematic difference in the location of the peak vertex between planning and execution across subjects (M1: $T^2_{2,20}$ = 0.725, p = 0.712; S1: $T^2_{2,20}$ = 2.424, p = 0.335).

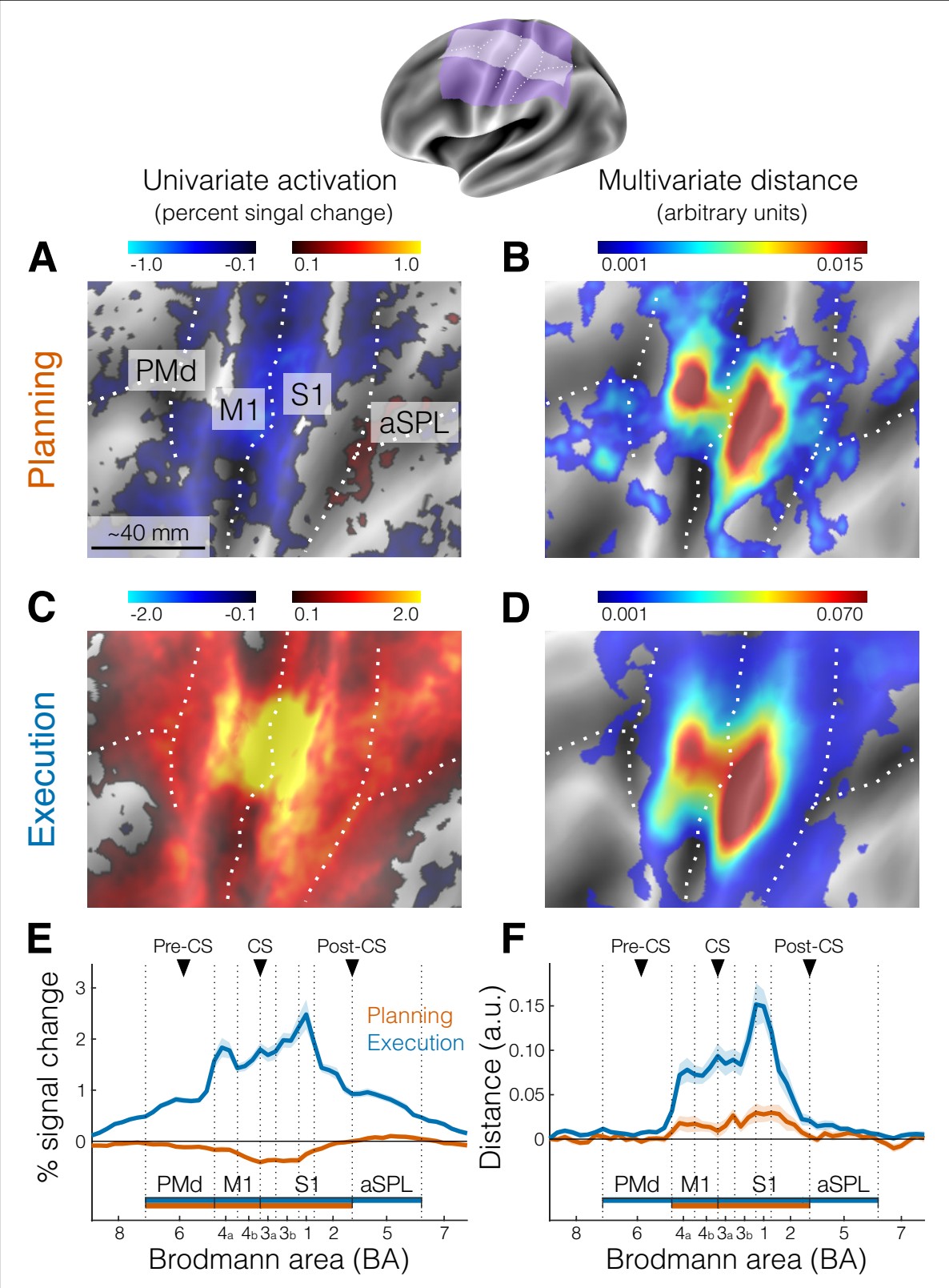

**Figure 2.** Activation and distance analyses of movement planning and execution. The inset shows the inflated cortical surface of the contralateral (left) hemisphere, highlighting the area of interest (A-D, purple) and the strip used for the profile region of interest (ROI) analysis (E, F, white). Major sulci are indicated by white dotted lines.( **A**) Univariate activation map (percent signal change) for the contrast planning > baseline (no-go trials only). (**B**) Multivariate searchlight map of the mean crossnobis distance between the planning of the three fingers (no-go trials only). (**C**) Same as A, but for the

*Figure 2 continued on next page*

*Figure 2 continued*

univariate contrast execution > baseline (go trials). (**D**) Same as B, but for the mean crossnobis distance between fingers during execution. Colorbars in A and C reflect mean percent signal change, whereas colorbars in B and D reflect mean crossnobis distance (arbitrary units). (**E**) Profile ROI analysis (see Materials and methods) of the mean percent signal change (± standard error of the mean [SEM]) during planning (no-go trials, orange) and execution (blue). The *x*-axis corresponds to Brodmann areas (BAs) selected from the white strip shown in the inset at the top. Horizontal bars indicate significance ($p < 0.05$) in a two-sided one-sample *t*-test vs zero for selected ROIs. (**F**) Same as E, but for the mean crossnobis distance (± SEM). Vertical dotted lines mark the approximate boundaries between BAs subdivisions of our main ROIs (see Materials and methods). Black triangles point to the approximate location of the main anatomical landmarks: Pre-CS = precentral sulcus; CS = central sulcus; Post-CS = postcentral sulcus. PMd (BA 6) = dorsal premotor cortex; M1 (BA 4a, 4b) = primary motor cortex; S1 (BA 3a, 3b, 1, 2) = primary somatosensory cortex; aSPL (BA 5) = anterior superior parietal lobule. For analogous results using the estimates of planning activity from all trials, see *Figure 2—figure supplement 1*. For the whole-brain maps of univariate and multivariate results, see *Figure 2—figure supplement 2*.

The online version of this article includes the following figure supplement(s) for figure 2:

**Figure supplement 1.** Activation and distance analyses using planning of both go and no-go trials.

**Figure supplement 2.** Whole-brain flat surface maps (both cortical hemispheres).

Together, our analyses indicate that information about single finger actions is already represented during motor planning in the same parts of the primary motor and somatosensory cortices that are engaged during execution of the presses. Given that we only used the activity estimates from no-go trials (~40% of total trials), this information cannot be explained by a spillover from subsequent execution-related activity. An analysis using the estimates of planning activity from all trials yielded very similar results (see *Figure 2—figure supplement 1*), demonstrating that we could separate planning from execution-related signals.

## Activity patterns are not caused by small movements during the preparation phase

The presence of planning-related information in primary sensorimotor regions was surprising, especially in S1, where it had not previously been reported in comparable fMRI studies (*Gallivan et al., 2016*; *Gallivan et al., 2011b*). To ensure that these results were not caused by overt movement, participants were instructed to maintain a steady force on the keyboard during the preparation phase, such that we could monitor even the smallest involuntary preparatory movements.

Inspection of the average force profiles (*Figure 3A*) revealed that participants were successful in maintaining a stable force between 0.2 and 0.4 N during preparation. However, averaging forces across trials may obscure small, idiosyncratic patterns visible during individual trials (*Figure 3B*) that could be used to distinguish the different movements. To test for the presence of such patterns, we submitted both the mean and standard deviation of the force traces on each finger to a multivariate dissimilarity analysis (see Materials and methods). Indeed, this sensitive analysis revealed that some participants showed small movement patterns predictive of the upcoming finger (positive behavioral distances in *Figure 3C*).

These distances, however, were ~200–300 times smaller than the average distances during execution (*x*-axis in *Figure 3D*), and we found no significant correlation between the magnitude of the behavioral differences for the preparation phase and the amount of planning information present in our sensory–motor ROIs (both p values for the slope of the linear fit >0.3 in *Figure 3C*). More importantly, a significantly positive intercept in the linear fit in *Figure 3C* (M1: p = 0.032; S1: p = 0.007) shows that, even after correcting for the influence behavioral patterns, the activity patterns in M1 and S1 remained informative (i.e., significantly positive neural distance even with no significant behavioral distance). Thus, the finding of finger-specific activity patterns in M1 and S1 cannot be explained by small involuntary movements during the preparation phase.

## Single finger activity patterns from planning to execution are positively correlated

How do planning-related activity patterns in M1 and S1 relate to the activity patterns observed during execution? Neurophysiological experiments have suggested that patterns of movement preparation

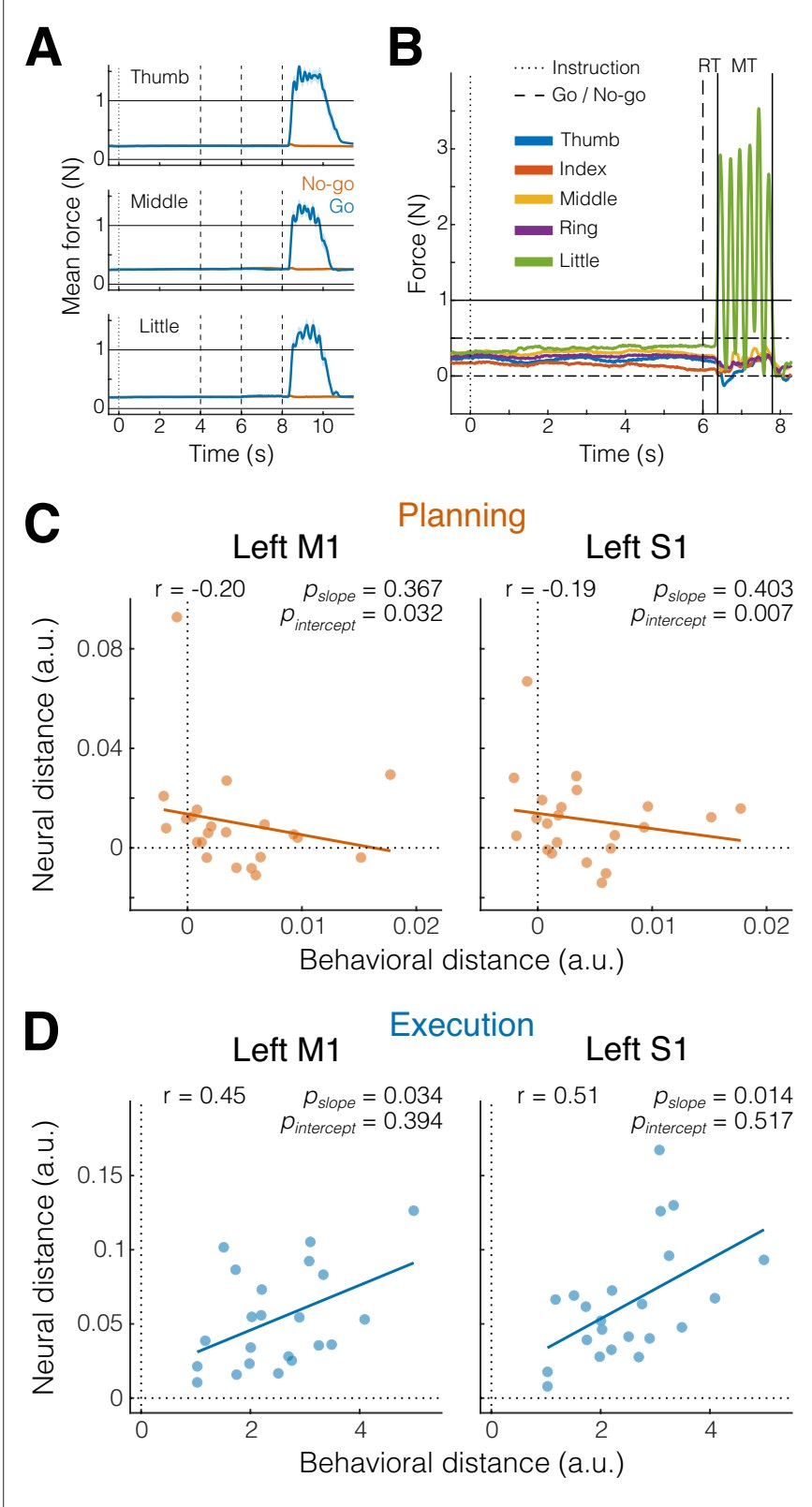

**Figure 3.** Small involuntary movements do not explain preparatory activity patterns in M1 and S1 (A). Mean finger force (± standard error of the mean [SEM]) plotted in 10 ms bins, time aligned to instruction onset (dotted vertical line) and end of the preparation phase (dashed vertical lines), separately for the three fingers and go (blue) and no-go (orange) trials. (**B**) Example of an individual trial with a 6-s preparation phase, followed six presses of the

*Figure 3 continued on next page*

*Figure 3 continued*

little finger (green). Horizontal solid line denotes press threshold (1 N). Dash-dotted lines denote the boundaries of the finger preactivation red area in *Figure 1A* (see Materials and methods). Reaction time (RT) was defined as the time from the go cue (dashed vertical line) to the onset of the first press (left solid vertical line). Movement time (MT) was defined as the time from the onset of the first press (left solid vertical line) until the release of the last press (right solid vertical line). (**C**) Pearson's correlation (*r*) between behavioral and neural distances in M1 and S1 (see Materials and methods) during the preparation phase (planning, orange). Each dot represents an individual participant (*N* = 22). Solid line shows linear regression fit; p values pairs refers to the slope and the intercept of the fitted line. (**D**) Same as C, but during the movement phase (execution, blue).

are orthogonal – or uncorrelated – to the patterns underlying active movement (*Kaufman et al., 2014*). This arrangement allows movement preparation to occur without causing overt movement.

When we compared the planning- and execution-related activity patterns as measured with fMRI, a technique that samples neuronal activity at a much coarser spatial resolution, we found the opposite result. Planning- and execution-related patterns for the same finger were tightly related. This can be seen already in the representational dissimilarity matrices (RDMs) that show the dissimilarity (crossnobis distance) for each pair of conditions (i.e., fingers 1 = thumb, 3 = middle, 5 = little for planning and execution phases).

First, the RDMs for M1 and S1 (*Figure 4A*) show a large difference between planning and execution patterns, which is due to the substantially higher average activation during movement compared to planning. This overall distance between planning and execution can also be appreciated in a three-dimensional (3D) projection of the RDMs using multidimensional scaling (MDS) to highlight the representational geometry between activity patterns (principal component PC1 in *Figure 4B*, *top*).

Second, within each phase, the pattern for the thumb was more distinct than those for the other fingers, replicating previous results from execution alone (*Ejaz et al., 2015*; *Yokoi et al., 2018*). Importantly, however, when ignoring the overall difference between the mean patterns for planning and execution, by looking at a rotated view of the representational geometry (*Figure 4B*, *bottom*), it became clear that the finger patterns were arranged in a congruent way, with planning- and execution-related activity patterns for the same finger being closer to one another. This representation suggested that the finger-specific patterns during planning may be a scaled-down version of the patterns during execution.

To test this idea more precisely, we quantified the correspondence (i.e., correlation) between planning and execution patterns for each finger using pattern component modeling (PCM, *Diedrichsen et al., 2018*). Because of the biasing influence of measurement noise, simple correlations between measured fMRI patterns are substantially lower than their true correlation (see http://www.diedrichsenlab.org/BrainDataScience/noisy_correlation). PCM corrects for this bias by evaluating the likelihood of the data (taking into account the measurement noise), under a range of models with a true correlation between 0 and 1. In other words, rather than asking which correlation value is the best estimate given the data, PCM asks how likely the data is given different correlation values (see Materials and methods for details).

The log-likelihood of the data under each model (evaluated individually for each participant and then averaged) is shown in *Figure 4C*. Across participants, the averaged maximum likelihood estimate of the correlation (i.e., the average best fitting correlation model) was *r* = 0.83 (±0.053 standard error of the mean [SEM]) for M1 and *r* = 0.81 (±0.061 SEM) for S1 (*Figure 4C*, red dashed lines). By comparing these estimates to the zero-correlation model, we can conclude that the correlation of finger-specific patterns across planning and execution was significantly larger than zero in both M1 and S1 (both $t_{21} > 13.288$, p < 1.086e−10 in a two-tailed one-sample *t*-test against zero). However, the maximum likelihood estimates of the correlation cannot be used to evaluate whether the overlap of these patterns was only partial (*r* < 1) or complete (*r* = 1), as the estimates are still biased due to measurement noise (*Walther et al., 2016*; *Diedrichsen et al., 2018*).

Therefore, in a cross-validated fashion, we compared for each participant the log-likelihood of the best fitting model (determined on all other participants, see Materials and methods) to the log-likelihood under the model that the patterns are perfectly correlated (*r* = 1). Across participants, this difference was not significant for either M1 ($t_{21}$ = 0.953, p = 0.176) or S1 ($t_{21}$ = 0.148, p = 0.442). Given that no correlation model had significantly higher log-likelihoods than the 1-correlation model, we

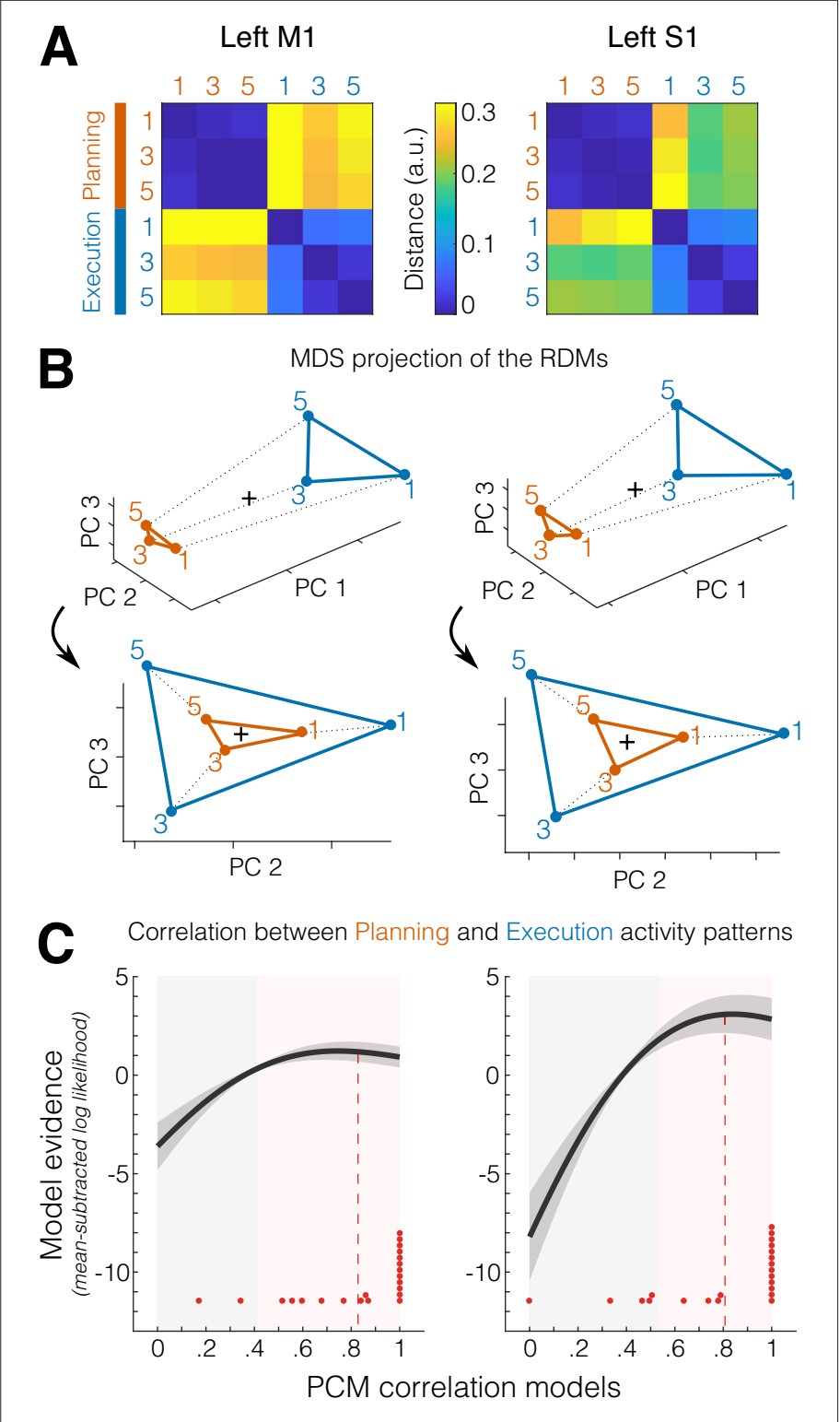

**Figure 4.** Correlated representations of single fingers across planning and execution.
(**A**) Representational dissimilarity matrices (RDMs) showing the average crossnobis distance between the activity patterns for digits 1 (thumb), 3 (middle), and 5 (little) during the preparation (no-go planning, orange) and movement (execution, blue) phases, for M1 (*left*) and S1(*right*), in the left hemisphere. (**B**) Two different views of a multidimensional scaling (MDS) plot that represents the distance between activity patterns in A as spatial distance in a three-dimensional (3D) coordinate system. *Top*, view highlighting the first principal component

*Figure 4 continued on next page*

*Figure 4 continued*

(PC1, difference in average activation between planning and execution). *Bottom*, rotated view highlighting the correspondence between representational geometries across planning and execution visible on PC2 and PC3. The black cross denotes the mean pattern across conditions. (**C**) Pattern component modeling (PCM) evaluation of models (*x*-axis) of different correlations between planning- and execution-related activity patterns. Shown in dark gray is the group average of the individual log-likelihood (± standard error of the mean [SEM] across participants) curves expressed as a difference from the mean log-likelihood across models (i.e., zero on the *y*-axis). Red dots indicate the best fitting correlation model for each participant (*N* = 22). Red dashed lines denote the average winning (i.e., best fitting) models across participants. Gray-shaded areas indicate models that perform statistically worse (p < 0.05) than the best fitting correlation model (determined in a cross-validated fashion, see Materials and methods). Pink-shaded areas indicate models that do not perform significantly worse than the best fitting correlation model (p ≥ 0.05).

cannot rule out that the underlying true correlation was indeed 1. In other words, we have as much evidence that the correspondence was only partial as we do that the correspondence was perfect. By comparing the best fitting correlation model to every other correlation model, we have evidence that the true (i.e., noiseless) correlation between planning and execution finger-specific activity pattern was between 0.41 and 1.0 in M1 and between 0.54 and 1.0 in S1 (*Figure 4C*, pink-shaded areas).

Thus, our data are consistent with the idea that, at the resolution of fMRI, the activity patterns for planning and execution of finger presses in S1 and M1 are either partially overlapping or even a scaled version of each other.

## Discussion

In the present study, we asked participants to produce repeated single finger presses while undergoing 7T fMRI. We used variable preparatory delays and no-go trials to cleanly dissociate the brain responses to the consecutive preparation and movement phases. We found that information about planned finger actions is present in both S1 and M1 before action onset, even though the overall level of activation in these regions was below resting baseline. Moreover, while execution elicited much higher brain activation, the fine-grained, finger-specific activity patterns were highly similar across planning and execution. Control analyses confirmed that the observed results were not caused by premovement finger activity.

Our finding that motor planning activates M1 in a finger-specific fashion was not necessarily surprising given many neurophysiological studies reporting anticipatory activity of M1 neurons related to movement intentions (*Tanji and Evarts, 1976*; *Riehle and Requin, 1989*; *Alexander and Crutcher, 1990*), as well as human neuroimaging showing shared information between delayed and immediate movement plans (*Ariani et al., 2018*). In contrast, the robust activity patterns related to single finger planning in S1 were more surprising, given that this region has classically been associated with the passive processing of somatosensory information from receptors in the skin, muscles, and tendons.

So, what could then be the role of S1 during movement planning? First, it is worth noting that there are substantial projections from S1 (BA 3a) that terminate in the ventral horn of the corticospinal tract (*Coulter and Jones, 1977*; *Rathelot and Strick, 2006*). Although stimulation of area 3a in macaques typically fails to evoke overt movements (*Widener and Cheney, 1997*), it has been suggested that this population of corticomotoneurons specifically projects to gamma motoneurons that control the sensitivity of muscle spindle afferents (*Rathelot and Strick, 2006*). Thus, it is possible that S1 plays an active role in movement generation by preparing the spindle apparatus in advance of the movement.

Second, the finger-specific preparatory state in S1 may reflect the prediction of the upcoming sensory stimulation, allowing for a movement-specific sensory gain control (*Azim and Seki, 2019*). It is likely that, this process is also accompanied by an allocation of attention to the cued finger. However, as voluntary (*Gallivan et al., 2011a*; *Gallivan et al., 2011b*) planning requires attention, our current dataset cannot distinguish between the two possibilities. Sensory stimuli could become attenuated to maintain movement stability and filter out irrelevant or self-generated signals. Indeed, multiple studies have shown that both somatosensation and somatosensory-evoked potentials in S1 decrease during voluntary movement (*Starr and Cohen, 1985*; *Chapman et al., 1987*; *Jiang et al., 1990*; *Seki and Fetz, 2012*). Alternatively, sensory processing of the expected salient signals could be enhanced to improve movement execution.

While several previous fMRI studies did not find action-specific encoding in S1 during planning (*Gallivan et al., 2011a*; *Gallivan et al., 2011b*, *Gallivan et al., 2016*; *Gallivan et al., 2013*), concurrently with our study a second paper found movement-specific modulation of S1 preparatory activity (*Gale et al., 2021*). Together, these two papers provide convergent evidence that motor planning triggers notable changes in the neural state of the somatosensory system and that such changes can be detected with fMRI in humans.

The second important finding in our paper was the close correspondence between finger-specific activity patterns across planning and execution – which appears to be at odds with the idea that these two processes occupy orthogonal neural subspaces to avoid overt movement during planning (*Kaufman et al., 2014*; *Elsayed et al., 2016*). We think that there are at least two possible explanations for this. First, the divergence of results could be caused by the difference in behavioral paradigms. While the neuronal correlates of movement planning in nonhuman primates have largely been studied using upper limb movements, we used here individuated finger presses. If for single finger actions even single-neuron activity patterns are highly correlated between planning and execution, then overt movement during planning would need to be actively suppressed, for example through the deactivation that we observed around the central sulcus.

An alternative and perhaps more likely explanation of the discrepancy lies in the different measurement modalities. Orthogonality was observed in electrophysiological recordings of individual neurons, whereas the fMRI measurements we employed here mainly reflect excitatory postsynaptic potentials (*Logothetis et al., 2001*) and average metabolic activity across hundreds of thousands of cortical neurons. Thus, it is possible that planning preactivates the specific cortical columns in M1 and S1 that are also most active during movement of that finger. Within each of these cortical microcircuits, however, planning-related activity could still be orthogonal to the activity observed during execution at the single-neuron level (e.g., see *Arbuckle et al., 2020*, for a similar observation for cortical representations of flexion and extension finger movements). This would suggest a new hypothesis for the architecture of the sensory–motor system where movement planning preactivates the action-specific circuits in M1 and S1. However, it does so in a fashion that the induced planning-related activity is, in terms of the firing output of neurons, orthogonal to the patterns during execution.

## Materials and methods

### Participants

Twenty-three right-handed neurologically healthy participants volunteered to take part in the experiment (13 F, 10 M; age 20–31 years, mean 23.43 years, SD 4.08 years). Criteria for inclusion were right-handedness and no prior history of psychiatric or neurological disorders. Handedness was assessed with the Edinburgh Handedness Inventory (mean 82.83, SD 9.75). All experimental procedures were approved by the Research Ethics Committee at Western University (HSREB 107061). Participants provided written informed consent to procedures and data usage and received monetary compensation for their participation. One participant withdrew before study completion and was excluded from data analysis (final $N$ = 22).

### Apparatus

Repeated right-hand finger presses were performed on a custom-made MRI-compatible keyboard device (*Figure 1A*). Participants only used the tips of their fingers to press on the keys. The keys of the device did not move but force transducers underneath each key measured isometric force production at an update rate of 2 ms (Honeywell FS series; dynamic range 0–25 N; sampling 200 Hz). A keypress/release was detected when the force crossed a threshold of 1 N. The forces measured from the keyboard were low pass filtered to reduce noise induced by the MRI environment, amplified, and sent to PC for online task control and data recording.

### Task

We used a task in which participants produced repeated keypresses with the tip of their right-hand fingers in response to numerical cues appearing on a computer screen (white outline, *Figure 1A*). On each trial, a string of six numbers (instructing cue) instructed which finger press to plan (1 = thumb, 3 = middle, 5 = little).

The length of the preparation phase (yellow background in *Figure 1*) was randomly sampled to be 4 s (56% of trials), 6 s (30%), or 8 s (14%). To limit and monitor unwanted movements during the preparation phase, we required participants to preactivate their fingers by maintaining a steady force of around 0.2–0.3 N on all of the keys during the preparation phase. As a visual aid, we displayed a red area (between 0 and 0.5 N) and asked participants to remain in the middle of that range with all the fingers (touching either boundary of the red area would count as unwanted movement, thus incurring an error). We preferred this technique over using electromyography (EMG) recordings to monitor micromovements during the preparation phase because extensive pilot experiments for our studies of ipsilateral representations (*Diedrichsen et al., 2013*) and mirroring (*Ejaz et al., 2018*) showed that force fluctuations from preactivated hand muscles provide a more sensitive readout of involuntary muscle activations compared to EMG signals acquired during fMRI.

At the onset of the movement phase (green background), participants received a color cue (go/no-go cue) indicating whether to perform the planned finger presses (blue outline = go, p = 0.6), or not (orange outline = no-go, p = 0.4). The role of no-go trials was to decouple the hemodynamic response to the successive planning and execution events, which would otherwise always overlap in fast fMRI designs due to the sluggishness of the fMRI response (*Ariani et al., 2018*).

To encourage planning during the delay period, at the go cue the digits were masked with asterisks, and participants had to perform the presses from memory. Participants had 2.5 s to complete the movement phase, and a vanishing white bar under the asterisks indicated how much time was left to complete all of the keypresses. Participants received online feedback on the correctness of each press with asterisks turning either green, for a correct press, or red, for incorrect presses. As long as the participants remained within task constraints (i.e., six keypresses in less than 2.5 s), an exact movement speed was not enforced. In no-go trials, participants were instructed to remain as still as possible maintaining the finger preactivation until the end of the movement phase (i.e., releasing any of the keys would incur an error).

During the reward phase (0.5 s, purple background) points were awarded based on performance and according to the following scheme: −1 point in case of no-go error or go cue anticipation (timing errors); 0 points for pressing any wrong key (press error); 1 point in case of a correct no-go trial; and 2 points in case of a correct go trial.

Intertrial intervals (ITI, gray background) were randomly drawn from {1, 2, 4, 8, 16 s} with the respective proportion of trials {0.52, 0.26, 0.13, 0.6, 0.3}.

## Experiment design and structure

Our chosen distribution of preparation times, ITIs, and no-go trials, was determined by minimizing the variance inflation factor (VIF) for a given length of scan:

$$VIF = \frac{var(E)}{var(X)}$$

where **var(E)** is the mean estimation variance of all the regression weights (planning- and execution-related regressors for each finger), and **var(X)** the mean estimation variance had these regressors been estimated in isolation. The VIF quantifies the severity of multicollinearity between model regressors by providing an index of how much the variance of an estimated regression coefficient is increased because of collinearity. Large values for VIF mean that model regressors are not independent of each other, whereas a VIF of 1 means no inflation of variance. After optimizing the design, the VIF was quite low, on average around 1.15, indicating that we could separate planning- and execution-related activity without a large loss of experimental power.

Participants underwent one fMRI session consisting of 10 functional runs and 1 anatomical scan. In an event-related design, we randomly interleaved three types of repeated single finger presses involving the tip of the thumb (1), the middle (3), and the little (5) fingers (e.g., 111,111 for thumb presses, *Figure 1A*) and three types of multifinger sequences (e.g., 135,315).

The day before the fMRI scan, participants familiarized themselves with the experimental apparatus and the go/no-go paradigm in a short behavioral session of practice outside the scanner (five blocks, about 15–30 min in total). This short training made the requirement of maintaining a steady force on all keys during the preparation phase very easy. In fact, the system was calibrated so that the natural weight of the hand on the keys was enough to bring the finger forces within the correct range

to be maintained. Thus, it is likely that little online control was required by the participants before pressing the keys.

For the behavioral practice, ITIs were kept to a fixed 1 s to speed up the task, and participants were presented with different sequences from what they would see while in the scanner. These six-item sequences were randomly selected from a pool of all possible permutations of the numbers 1, 3, and 5, with the exclusion of sequences that contained consecutive repetitions of the same number. Given that the current paper is concerned with the relationship between representations of simple planning and execution, here we will focus only on the results for single finger actions. The results for multifinger sequences are intended for publication in a future paper.

Each single finger trial type (e.g., 111,111) was repeated five times (two no-go and three go trials), totaling 30 trials per functional run. Two periods of 10 s rests were added at the beginning and at the end of each functional run to allow for signal relaxation and provide a better estimate of baseline activation. Each of the 10 functional runs took about 5.5 min and the entire scanning session (including the anatomical scan and setup time) lasted for about 75 min.

## Imaging data acquisition

High-field fMRI data were acquired on a 7T Siemens Magnetom scanner with a 32-channel head coil at Western University (London, Ontario, Canada). The anatomical T1-weighted scan of each participant was acquired halfway through the scanning session (after the first five functional runs) using a Magnetization-Prepared Rapid Gradient Echo sequence (MPRAGE) with voxel size of 0.75 × 0.75 × 0.75 mm isotropic (field of view = 208 × 157 × 110 mm [A–P, R–L, F–H], encoding direction coronal). To measure the blood-oxygen-level-dependent responses in human participants, each functional scan (330 volumes) used the following sequence parameters: GRAPPA 3, multiband acceleration factor 2, repetition time (TR) = 1.0 s, echo time (TE) = 20 ms, flip angle (FA) = 30°, slice number: 44, voxel size: 2 × 2 × 2 mm isotropic. To estimate and correct for magnetic field inhomogeneities, we also acquired a gradient echo field map with the following parameters: transversal orientation, field of view: 210 × 210 × 160 mm, 64 slices, 2.5 mm thickness, TR = 475 ms, TE = 4.08 ms, FA = 35°.

## Preprocessing and univariate analysis

Preprocessing of the functional data was performed using SPM12 (fil.ion.ucl.ac.uk/spm) and custom MATLAB code. This included correction for geometric distortions using the gradient echo field map (*Hutton et al., 2002*), and motion realignment to the first scan in the first run (three translations: *x*, *y*, *z*; three rotations: pitch, roll yaw). Due to the short TR, no slice timing corrections were applied. The functional data were coregistered to the anatomical scan, but no normalization to a standard template or smoothing was applied. To allow magnetization to reach equilibrium, the first four volumes of each functional run were discarded. The preprocessed images were analyzed with a general linear model (GLM). We defined separate regressors for each combination of the six finger actions (single, multi) × two phases (preparation, movement). To control for the effect of potential overlap between execution activity and the preceding planning activity, we also estimated a separate GLM with separate regressors for the preparation phases of go and no-go trials, resulting in a total of 18 regressors (12 go + 6 no-go), plus the intercept, for each run. Each regressor consisted of a boxcar function (on for 2 s of each phase duration and off otherwise) convolved with a two-gamma canonical hemodynamic response function with a peak onset at 5 s and a poststimulus undershoot minimum at 10 s (*Figure 1B*).

Given the relatively low error rates (i.e., number of error trials over total number of trials, timing errors: 7.58 ± 0. 62%; press errors: 1.18 ± 0.26%, see Task), all trials were included to estimate the regression coefficients, regardless of whether the execution was correct or erroneous. Ultimately, the first-level analysis resulted in activation images (beta maps) for each of the 18 conditions per run, for each of the participants.

## Surface reconstruction and ROI definition

Individual subject's cortical surfaces were reconstructed using Freesurfer (*Dale et al., 1999*). First, we extracted the white-gray matter and pial surfaces from each participant's anatomical image. Next, we inflated each surface into a sphere and aligned it using sulcal depth and curvature information to the Freesurfer average atlas (*Fischl et al., 1999*). Both hemispheres in each participant were then

resampled into Workbench's 164 k vertex grid. This allowed us to compare similar areas of the cortical surface in each participant by selecting the corresponding vertices on the group atlas.

Anatomical ROIs were defined using a probabilistic cytoarchitectonic atlas (*Fischl et al., 2008*) projected onto the common group surface. Our main ROIs were defined bilaterally as follows: primary motor cortex (M1) was defined by including nodes with the highest probability of belonging to BAa 4a and 4b, within 2 cm above and below the hand knob anatomical landmark (*Yousry et al., 1997*); primary somatosensory cortex (S1) was defined by the nodes related to BA 1, 2, 3a, and 3b; PMd was defined at the junction between the superior frontal sulcus and the precentral sulcus (BA 6); finally, the anterior part of the superior parietal lobule (aSPL, BA 5) included areas anterior, superior, and ventral to the intraparietal sulcus. ROI definition was carried out separately in each subject using FSL's subcortical segmentation. When resampling functional onto the surface, to avoid contamination between M1 and S1 activities, we excluded voxels with more than 25% of their volume in the gray matter on the opposite side of the central sulcus.

## Multivariate distance analysis

To detect single finger representations across the cortical surface, we used representational similarity analysis (RSA; *Diedrichsen and Kriegeskorte, 2017*; *Walther et al., 2016*) with a surface-based searchlight approach (*Oosterhof et al., 2011*). For each node, we selected a region (the searchlight) corresponding to 100 voxels (12 mm disc radius) in the gray matter and computed cross-validated Mahalanobis (crossnobis, *Walther et al., 2016*) dissimilarities between pairs of evoked activity patterns (beta estimates from first-level GLM) of single finger sequences, during both preparation and movement phases.

Prior to calculating the dissimilarities, beta weights for each condition were spatially prewhitened that is weighted by the matrix square root of the noise covariance matrix estimated from the residuals of the GLM. The noise covariance matrix was slightly regularized toward a diagonal matrix (*Ledoit and Wolf, 2004*). Multivariate prewhitening has been shown to increase the reliability of dissimilarity estimates (*Walther et al., 2016*). The dissimilarity was then computed by multiplying the difference between two conditions patterns with the pattern difference for the same conditions of any other run, and then averaging over all runs and voxels. This resulted in 15 dissimilarities between the 6 conditions (3 single fingers, separately for planning and execution), which can be visualized as a 6 × 6 RDM (*Figure 4A*). An alternative visualization can be obtained using classical MDS, which shows the six conditions as points projected into the 3D space spanned by the eigenvectors of the patterns that were associated with the three largest eigenvalues (i.e., the three principal components).

For the searchlight analysis, we assigned the average distance between any of the three planning conditions to the central node of the searchlight. The region was then moved across all nodes across the surface sheet obtaining a cortical map (*Figure 2B*). An equivalent analysis was conducted for the execution patterns (*Figure 2D*). Cross-validation ensures the distances estimates are unbiased, such that if two patterns differ only by measurement noise, the mean of the estimated value would be zero. This also means that estimates can sometimes become negative. Therefore, dissimilarities significantly larger than zero indicate that two patterns are reliably distinct, similar to an above-chance performance in a cross-validated pattern-classification analysis.

The searchlight analysis was mainly used for visualization purposes. Additionally, we conducted the multivariate analysis separately for each anatomically defined ROI (e.g., *Figure 4A*). For the profile ROI analysis (both univariate and multivariate, e.g., *Figure 2E, F*), we defined 50 rectangular surface-based searchlights in each hemisphere that covered the virtual strip shown in the top inset of *Figure 2* and that were aligned to the boundaries between different ROI subdivisions. Based on these surface-based searchlights, we defined the voxel-based subdivisions in individual brains. For statistical comparisons, these subdivisions were successively grouped by averaging within-ROI subdivisions (see *Figure 2E, F*). This approach allowed us to compute both ROI-level statistical comparisons and the analysis of the ratio of distances in the different subdivisions of our main ROIs (e.g., M1 into BA4a and BA4b). Statistical comparisons consisted of two-sided one-sample *t*-test vs zero for selected ROIs.

## Correlation between behavioral and neural distances

To ensure that our planning results were not contaminated by unwanted micromovements during the preparation phase, we calculated the behavioral distance between the different fingers on the basis of keyboard force data and correlated behavioral and neural distances.

For behavioral distances, we first extracted force data (2-ms temporal resolution, smoothed with a Gaussian kernel of 9.42 full width at half maximum, FWHM) and binned it in 10 ms steps (downsampling largely due to memory constraints) for both the preparation and movement phases (*Figure 3A*). Next, for each subject, we calculated the mean (5) and the standard deviation (5) of the time-averaged force of each finger for each condition (3 sequences × 2 phases = 6) and block (10). These subject-specific finger force patterns (60 × 10) were multivariately prewhitened using their covariance matrix. Finally, we calculated the cross-validated squared Euclidean distances for each condition (6 × 6 RDM) and averaged distances between the three finger presses for each phase (preparation, movement).

These mean finger force distances for each subject were correlated with the mean voxel activity distances from the two phases for two ROIs (M1 and S1, *Figure 3C,D*). To statistically assess that the neural distances were still significantly larger than zero even in the absence of behavioral distances, we computed the pvalue for the intercept of the linear fit.

## PCM correlation models

Visual inspection of the RDM and MDS plots (*Figure 4A, B*) suggested that the finger-specific activity patterns during planning and execution might be arranged in a congruent fashion. This correspondence can be assessed by determining the correlation between the planning and execution activity patterns for matching fingers (i.e., planning finger 1 with executing finger 1), after accounting for the average activity pattern for planning and execution across fingers.

The problem with simple Pearson's correlations or cross-validated correlations is that these measures are biased by noise. Even if the patterns for planning and execution were perfectly correlated (i.e., a scaled version of each other), the empirical correlation estimates would not be one (see http://www.diedrichsenlab.org/BrainDataScience/noisy_correlation). Therefore, we used (PCM *Diedrichsen et al., 2018*, openly available at github.com/DiedrichsenLab/PcmPy; copy archived at swh:1:rev:076b9a685ed116b1f0b83a68a0955d0cc5323a42, *Ariani, 2022*) to generate different models, each assuming a specific correlation between planning and execution patterns on the interval between 0 and 1 in steps of 0.01. We then computed the log-likelihood of the observed data ($Y$, the 6 activation patterns observed in 10 runs) from each participant under each correlation model ($r$): $p(Y|r)$, which is plotted in *Figure 4C*. PCM assumes that both the true activity patterns and the measurement noise are randomly distributed with a multivariate Gaussian distribution. The likelihood of the data under each model can then be analytically evaluated (for details see *Diedrichsen et al., 2011*; *Diedrichsen et al., 2018*; *Diedrichsen et al., 2021*). This likelihood depends only on the covariance matrix of the measured activity patterns and the predicted covariance matrix from the model. (Note: more precisely it relies on the measured and the predicted second moment matrix, as we do not subtract out the mean of each pattern across voxels.) Like RSA, PCM therefore abstracts away from the actual activity patterns, as it only depends on the relationship between the patterns, but does not have to model the pattern themselves. In fact, there is a 1:1 relationship between the second moment matrix used in PCM and the RDM used in RSA (*Diedrichsen and Kriegeskorte, 2017*). In both cases, the correlation between two conditions (e.g., planning and execution, for each finger) can be seen as the diagonal of the off-diagonal block of this matrix. The approach is also equivalent to an encoding model estimated with Ridge regression (*Diedrichsen and Kriegeskorte, 2017*), with the advantage that it can be estimated in closed form without fallback on cross-validation.

Here, we used 100 PCM correlation models with correlations in the range [0–1] in equal step sizes. The number of correlation models was chosen arbitrarily—ultimately, it only determines the amount of correlation values tested (i.e., the resolution on the *x*-axis in *Figure 4C*). By exploring the entire log-likelihood function across different correlations models, this approach allows us to test specific hypothesis even if the signal-to-noise level is low.

Apart from a fixed correlation, each model contained five free parameters, each describing the variance of specific pattern component. The first two parameters captured the variance of the common pattern for all execution patterns and the variance of the common pattern for all planning patterns. Together, these two components captured the overall difference between planning and execution.

The next two parameters captured the variance associated with the three fingers under the two conditions. Finally, the noise parameter determined the variance of the measurement noise. Because all correlation models had the same number of parameters, we simply maximized the likelihood for each correlation model in respect to these parameters.

The curve in *Figure 4C* shows the average log-likelihood for each correlation model (100 models from 0 to 1 in equal steps sizes), relative to the mean log-likelihood across models (zero on the *y*-axis). Differences between the log-likelihoods can be interpreted as log-Bayes factors. Group inferences were performed using a simple *t*-tests on the log-likelihoods.

To compare specific models to the best fitting model, we had to correct for the bias arising from picking the best model and testing it on the same data. Therefore, we used $N - 1$ subjects to determine the group winning model, and then chose the log-likelihood of this model for the left-out subject (for whom this model may not be the best one) as the likelihood for the 'best' model. This was repeated across all subjects and a one-sided paired-sample *t*-test was performed on the recorded log-likelihood and every other model.

This test revealed which of the correlation models were significantly worse (i.e., associated with a lower log-likelihood) than the winning model that was independently estimated via cross-validation (gray-shaded area in *Figure 4C*).

In sum, PCM has the advantage over alternative approaches in that it provides stable inferences even for noisy data, offering an optimal evaluation (in the likelihood sense) of the real evidence present in the data about the true correlation between two activity patterns. For technical implementation details of PCM, including a full example of PCM correlation models, see the documents of the openly available toolbox written in Python (pcm-toolbox-python.readthedocs.io/en/latest/demos/demo_correlation.html).

## Acknowledgements

This work was supported by a NSERC Discovery Grant (RGPIN-2016-04890) awarded to JD, and the Canada First Research Excellence Fund (BrainsCAN). The authors wish to thank Eva Berlot for helpful discussions and contributions to data analysis.

## Additional information

### Competing interests

J Andrew Pruszynski, Jörn Diedrichsen: Reviewing editor, *eLife*. The other author declares that no competing interests exist.

### Funding

| Funder | Grant reference number | Author |
| --- | --- | --- |
| Canada First Research Excellence Fund | BrainsCAN | Jörn Diedrichsen |

The funders had no role in study design, data collection, and interpretation, or the decision to submit the work for publication.

### Author contributions

Giacomo Ariani, Conceptualization, Data curation, Formal analysis, Investigation, Methodology, Project administration, Software, Validation, Visualization, Writing - original draft, Writing - review and editing; J Andrew Pruszynski, Supervision, Writing - review and editing; Jörn Diedrichsen, Conceptualization, Funding acquisition, Methodology, Project administration, Resources, Software, Supervision, Writing - review and editing

### Author ORCIDs

Giacomo Ariani http://orcid.org/0000-0001-9074-1272
J Andrew Pruszynski http://orcid.org/0000-0003-0786-0081
Jörn Diedrichsen http://orcid.org/0000-0003-0264-8532

### Ethics

All experimental procedures were approved by the Research Ethics Committee at Western University (HSREB protocol 107061). Participants provided written informed consent to procedures and data usage and received monetary compensation for their participation.

### Decision letter and Author response

Decision letter https://doi.org/10.7554/eLife.69517.sa1
Author response https://doi.org/10.7554/eLife.69517.sa2

## Additional files

### Supplementary files
• Transparent reporting form

### Data availability

The data used to create the figures in this study can be found on github https://github.com/g14r/single-finger-planning (copy archived at swh:1:rev:076b9a685ed116b1f0b83a68a0955d0cc5323a42).

The following dataset was generated:

| Author(s) | Year | Dataset title | Dataset URL | Database and Identifier |
|---|---|---|---|---|
| Ariani G, Pruszynski JA, Diedrichsen J | 2021 | Finger sequence planning | https://openneuro.org/datasets/ds003684 | OpenNeuro, 10.18112/openneuro.ds003684.v1.0.0 |

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
