## [Editor Report]

In this elegant and rigorous study, the authors investigated the neural correlates of planning and execution of single finger presses in a 7T fMRI study focusing on primary somatosensory (S1) and motor (M1) cortices. BOLD patterns of activation/deactivation and finger-specific pattern discriminability indicate that M1 and S1 are involved not only during execution, but also during planning of single finger presses. These important results clearly establish that the role of primary somatosensory cortex goes beyond pure processing of tactile information and will be of great interest for researchers in the field of motor control and of systems neuroscience.

---

## [Decision Letter]

**Decision letter after peer review:**

Thank you for submitting your article "Motor planning brings human primary somatosensory cortex into action-specific preparatory states" for consideration by *eLife*. Your article has been reviewed by 3 peer reviewers, and the evaluation has been overseen by Chris Baker as the Reviewing Editor/Senior Editor. The following individuals involved in review of your submission have agreed to reveal their identity: Andrea Serino (Reviewer #1); Sanne Kikkert (Reviewer #3).

Essential revisions:

1) Expand and clarify descriptions of the methods and analyses.

2) Conduct additional analyses to help substantiate the claim that planning and execution patterns are scaled version of each other.

3) Conduct additional analyses of SMA and pre-SMA to show the expected planning-related pattern of activity in these areas, especially given the null results in premotor and parietal areas.

4) Given another recent publication with related results (Gale et al., 2021), consider expanding the results to include ipsilateral regions, whole volume analyses (see Reviewer #2).

5) Temper interpretations of null results.

*Reviewer #1 (Recommendations for the authors):*

• To help the reader, the description of the PCM analyses should be improved. In particular:

1. It is not clear how the correlation models are conceived.

2. It is not clear under which form the data are treated (pattern, RDM or second-moment matrix) and how the planning/execution are compared under a given correlation model.

3. The overall procedure of PCM should be better described in the methods.

4. Figure 4D is rather hard to understand and the legends should be improved.

• Providing a comparison between M1 and S1 (at the RDM level) could be interesting (e.g., is the representational structure more similar between M1 and S1 for planning or for execution?).

*Reviewer #2 (Recommendations for the authors):*

Even though the study is of high quality, I have a concern regarding the fact that recent research showed similar evidence for the role of S1 during motor planning (Gale et al., 2021). These findings partially limit the novelty and the impact of the present investigation. Following the evidence emerging from this recent research, I think the authors might consider conducting additional analyses to strengthen their research and better characterise the presented results.

Given the high spatial resolution of the study, I think that it might be helpful to focus on specific sub-fields of the somatosensory cortex to dissect their specific roles. In addition, based on the recent description of the possible role of ipsilateral cortices in representing movements, it would be also interesting to investigate the role of ipsilateral S1 (and M1) and to compare the pattern of finger-specific movements within these regions with the regions of the contralateral hemisphere. Finally, it would be interesting to provide whole brain data both for the univariate and the multivariate analyses in order to provide a global overview of the involved brain regions and of the brain areas hosting finger-specific patterns.

1) Lack of activation during the planning phase

Most previous fMRI studies on motor planning using complex movements instructed participants to stay still during the planning phase. It is not surprising to see deactivation or activation at baseline during this phase.

In this study, the lack of activation in S1 and M1 during planning is surprising for me, given that the participants were engaged in a visuomotor task throughout the whole duration of the preparation/planning phase (Figure 1A, lines 303-308). Participants had to pre-activate their fingers by maintaining a steady force on all the keys during the preparation phase. Moreover, a visual feedback showing the applied force was displayed and participants had to apply a force to maintain the level of this feedback within a specific range.

This type of task is rather demanding, so I would expect to see the engagement of parieto-frontal motor networks during motor planning. Could it be possible that the lack of activation in S1 and M1 during planning is simply due to how the data were modelled in the GLM. The authors reported that they modelled only the first 2 seconds of the preparation phase (lines 394-397) as this phase has different durations (4-8 seconds). As an additional control analysis, it might be interesting to directly compare the preparation phase with the initial and final baseline by explicitly modelling the baseline in the GLM.

As a general suggestion, I would recommend the authors to provide whole brain analysis as supplementary materials (see point 4).

2) Specificity of the described effects in the somatosensory cortex

I would suggest investigating the role of the different subsections of S1. One possibility could be to adopt the parcellation of the Anatomy toolbox, as implemented in a recent investigation (Gale et al., 2021), or by analysing separately the different cytoarchitectonic maps adopted to create the S1 ROI (see lines 416-417). This study (Gale et al., 2021) suggest that S1 subsectors might represent different information during action planning. Support to the possible role of S1 in motor control comes from a recent neurophysiological study showing the possible encoding of complex arm movements, and not only of somatosensory information, within area BA 2 of macaque monkeys (Chowdhury et al., 2020). The higher spatial resolution of the present study might provide support to the data presented in the and/or new insights of the specificity of the described effect in the somatosensory cortex.

3) Role of ipsilateral hemisphere

I would also suggest conducting the same type of analyses on S1 and M1 of the ipsilateral hemisphere as a meaningful control site. Moreover, this analysis could provide novel insights on the possible role of ipsilateral S1 in representing ipsilateral movements. Indeed, there is a growing body of evidence showing the representation of ipsilateral movements in the motor cortex (Bundy and Leuthardt, 2019), but much less is known on the role of S1. If a representation of the planned movements is present in one or more subsectors of S1 (see Gale et al. 2021), the comparison between the informative patterns across the two phases of the task (and maybe even across hemispheres) might be also of interest.

4) Whole brain univariate analyses and searchlight analyses

I think it would be useful to provide whole brain data of the univariate and multivariate analyses for the two phases of the task. The authors might consider adding the results of these analyses as supplementary figures.

Whole brain univariate results could provide an overview of the network of regions engaged during the planning of finger movements. Looking at Figure 1, it seems that there is only an anterior parietal region recruited by the task (possibly at an uncorrected level). I would expect to see activation in parieto-frontal regions. This analysis might also partially answer to my first point.

I also would suggest conducting searchlight analyses for the two phases of the task. These analyses might serve different purpose. First, these results could provide insights on additional regions (e.g. parietal regions, SII) representing different planned finger movements. Second, as most (if not all) previous MVPA studies adopting paradigm with a delayed execution in the field investigated complex hand/arm actions, the present data might provide an indirect indication on which part of the parieto-frontal motor networks are engaged for simple finger movements and which are specific for more complex arm/hand actions.

Finally, the comparison/conjunction between univariate and multivariate results might provide insights on the regions within the motor networks which are both engaged by the task and host a representation of the different movements. This analysis would be particularly interesting for the planning phase of the task.

5) Comparison of the topography of informative patterns between the planning and execution phase of the task

I apologise in advance if this concern is related to my misinterpretation of this analysis, as I'm not familiar with the Hotelling T2 test (lines 130-132). My understanding of this test is that it allows to test the difference between two (multivariate) means of different distributions. I think that the authors compared the distribution of the crossnobis distance for M1 and S1 across subjects.

If my understanding of the analysis is correct, then this analysis showed that the distribution of crossnobis distances within M1 and S1 is consistent across the two phases, but this analysis doesn't directly test if the spatial organization of the informative patterns across the 2 phases is different. I would suggest to directly compare the distribution of crossnobis distance between the two phases across subjects independently in each vertex of M1 and S1. The result of this analysis would be a spatial map of the difference between the distributions of the two phases. This would allow to appreciate if there is any spatial asymmetry between the 2 phases.

As a general recommendation, I would suggest providing additional details in the description of this analysis to allow an easier understanding of its implications.

*Reviewer #3 (Recommendations for the authors):*

1. I urge the authors to include more details regarding the analysis they conducted. This is an overall recommendation, but a few things were specifically unclear to me:

a. Since the PCM is not a commonly used analysis, the manuscript would especially benefit from more conceptual explanation regarding the rationale of conducting this analysis, further explanation on the method, and how the achieved r-values can be interpreted.

b. What does the colourbar scaling represent in Figure 1B and C – is this percent signal change?

c. Similarly – Are all coloured vertices in Figure 2A-D significantly activated/ have significant distances? Or what does a value of 0.1 and 0.001 represent? Also, are the maps in Figure 2A and C corrected for multiple comparisons? This is not reported in the Methods section or the figure legend.

d. What analysis was ran to test for informative patterns in M1 and S1 while correcting for the influence of behavioural patterns – this analysis is not reported in the methods section.

e. Did the authors correct for multiple comparisons across the ROIs tested?

f. For the analysis presented Figs 2E and F, did the authors average across the y-coordinates on the whole brain flatmap, or were this values extracted from a single straight line? i.e. do the distances on the cortical surface represent averages or rather values from single vertices? This analysis is not mentioned in the Methods section.

g. The PCM rationale is better explained in the Results section, from only reading the methods section the rationale behind this method was unclear.

h. Regarding the PCM analysis: If a model of 0.4 correlation is most predictive, can we conclude the representational patterns are significantly correlated? Or how can we interpret these correlational models?

i. Regarding the PCM analysis: How could a best fitting model perform better from the one-correlation model (i.e. does it make sense to test achieved r>1)? Isn't a correlation of 1 the maximum achievable value? I think that if activity patterns during planning and execution are truly a mere scaled version of each other, the correlation between the activity patterns should be 1. So if the achieved r is sig less than 1, this would argue against the scaled version argument. This means that the authors should test if their model performs sig less than the one-correlation model.

2. What could potentially be added to the argument against micromovements explaining neural distances in S1/M1 during movement planning is that the expected correlation would be positive, while you see a non-significant negative correlation. It would also be interesting to test whether there are significant differences between the behavioural and neural distances during planning and execution for each ROI. For me that would be more convincing than showing a mere non-significant correlation (without Bayesian stats).

---

## [Author Response]

Reviewer #1 (Recommendations for the authors):• To help the reader, the description of the PCM analyses should be improved. In particular:1. It is not clear how the correlation models are conceived.

Each correlation model was created as a second moment matrix with a fixed correlation value on the diagonal of the off-diagonal blocks. The number of correlation models was chosen arbitrarily, in N steps between 0 and 1 (in our case N = 100). Ultimately, the number of models does not matter, it only determines the resolution of correlation values tested (i.e., the resolution on the x-axis in Figure 4C). We now provide more details in the description of PCM analyses (page 24, line 733-758).

2. It is not clear under which form the data are treated (pattern, RDM or second-moment matrix) and how the planning/execution are compared under a given correlation model.

In PCM a representational model is formulated by its predicted second moment matrix. The data is evaluated by determining the likelihood of the data under a Gaussian distribution with that second moment. We transformed the second moment matrices into RDMs in Figure 4A, as we believe that many readers will find an RDM slightly more intuitive. However, there is a 1:1 relationship between second moment matrices and RDMs. In both cases, the correlation between planning and execution (across fingers) can be seen as the diagonal of the off-diagonal block of this matrix.

3. The overall procedure of PCM should be better described in the methods.

We now expanded our description of the overall PCM procedure (page 23-24, line 717-758). In addition to the original PCM paper, we now also provide a link to a Jupyter Notebook example of running a PCM analysis for a simulated example with a very similar structure.

4. Figure 4D is rather hard to understand and the legends should be improved.

We have now added information to the figure legend to improve understanding of the different components in Figure 4.

• Providing a comparison between M1 and S1 (at the RDM level) could be interesting (e.g., is the representational structure more similar between M1 and S1 for planning or for execution?).

Because we measured only 3 fingers, the RDM within planning and execution is characterized only by three distances. Thus, a formal comparison of the RDMs between M1 and S1 has very little power. However, as can be seen from Figure 4B, the average RDM structure stays very comparable across planning and execution both in M1 and S1. Indeed, in a more complete experiment where we measured all 5 fingers (Ejaz et al., 2015, supplementary figure 4), we found no significant difference in the RDM structure across M1 and S1.

Reviewer #2 (Recommendations for the authors):Even though the study is of high quality, I have a concern regarding the fact that recent research showed similar evidence for the role of S1 during motor planning (Gale et al., 2021). These findings partially limit the novelty and the impact of the present investigation. Following the evidence emerging from this recent research, I think the authors might consider conducting additional analyses to strengthen their research and better characterise the presented results.Given the high spatial resolution of the study, I think that it might be helpful to focus on specific sub-fields of the somatosensory cortex to dissect their specific roles. In addition, based on the recent description of the possible role of ipsilateral cortices in representing movements, it would be also interesting to investigate the role of ipsilateral S1 (and M1) and to compare the pattern of finger-specific movements within these regions with the regions of the contralateral hemisphere. Finally, it would be interesting to provide whole brain data both for the univariate and the multivariate analyses in order to provide a global overview of the involved brain regions and of the brain areas hosting finger-specific patterns.

We thank Reviewer #2 for the overall appreciation of our study. We agree that adding whole-brain maps (including ipsilateral M1/S1) as supplementary material would provide a more complete picture of our results. Using 3 possible actions and advanced statistical methods like PCM to test for the correspondence of movement-specific activation patterns between planning and execution, our study provides an important further insight into the planning activity in M1 and S1. Second, on a technical note, we believe that our paper provides a more thorough control for micro-movements during planning (by monitoring finger forces during the preparation phase) and a cleaner separation of planning and execution (by limiting the analysis to no-go trials). We now acknowledge the Gale et al. (2021) paper in the Discussion section (page 16, line 456-461).

1) Lack of activation during the planning phaseMost previous fMRI studies on motor planning using complex movements instructed participants to stay still during the planning phase. It is not surprising to see deactivation or activation at baseline during this phase.In this study, the lack of activation in S1 and M1 during planning is surprising for me, given that the participants were engaged in a visuomotor task throughout the whole duration of the preparation/planning phase (Figure 1A, lines 303-308). Participants had to pre-activate their fingers by maintaining a steady force on all the keys during the preparation phase. Moreover, a visual feedback showing the applied force was displayed and participants had to apply a force to maintain the level of this feedback within a specific range.This type of task is rather demanding, so I would expect to see the engagement of parieto-frontal motor networks during motor planning. Could it be possible that the lack of activation in S1 and M1 during planning is simply due to how the data were modelled in the GLM. The authors reported that they modelled only the first 2 seconds of the preparation phase (lines 394-397) as this phase has different durations (4-8 seconds). As an additional control analysis, it might be interesting to directly compare the preparation phase with the initial and final baseline by explicitly modelling the baseline in the GLM.As a general suggestion, I would recommend the authors to provide whole brain analysis as supplementary materials (see point 4).

Reviewer #1 made a similar remark about the nature of the baseline task and the lack of activation during planning. In short, we have good reasons to believe that this can be explained by a combination of training effect and limited planning demands for single finger presses.

2) Specificity of the described effects in the somatosensory cortexI would suggest investigating the role of the different subsections of S1. One possibility could be to adopt the parcellation of the Anatomy toolbox, as implemented in a recent investigation (Gale et al., 2021), or by analysing separately the different cytoarchitectonic maps adopted to create the S1 ROI (see lines 416-417). This study (Gale et al., 2021) suggest that S1 subsectors might represent different information during action planning. Support to the possible role of S1 in motor control comes from a recent neurophysiological study showing the possible encoding of complex arm movements, and not only of somatosensory information, within area BA 2 of macaque monkeys (Chowdhury et al., 2020). The higher spatial resolution of the present study might provide support to the data presented in the and/or new insights of the specificity of the described effect in the somatosensory cortex.

Thanks for this suggestion, we agree that looking into different S1 subsections could potentially enrich our paper. Therefore, we have repeated our profile ROI analysis (Figure 2E-2F) to allow for separation of the different subdivisions of M1 and S1. We used rectangular searchlights (50 per virtual strip in each hemisphere) that were defined to align with the boundaries between the different subdivision (see Author response image 1, where each color corresponds to a different searchlight). Based on these surface-based ROIs, we defined the voxel-based ROIs in the individual brains. For statistical purposes, the rectangular searchlights were successively grouped by averaging within-ROI subdivisions (see Author response image 1). This surface-based approach provides a more accurate separation of different subdivisions than the volume-based approach—see Fischl et al. (2008). The subdivisions are now clearly labeled in the profile plots (Author response image 1) and new (Figure 2E-2F).

**Author response image 1. sa2fig1:** A. Example of rectangular searchlights used to examine different subsections of our ROIs, superimposed on the flat map of the left hemisphere. Each color corresponds to a different searchlight (i.e., one point on the x-axis in B). Black outlines denote approximate boundaries of the different subdivisions of the ROIs (see also vertical dotted lines in B). White dotted lines indicate the three main sulci for our regions of interest: precentral sulcus (pre-CS), central sulcus (CS), and postcentral sulcus (post-CS). B. New crossnobis distance analysis (top) and corresponding ratio of the distances (bottom) during planning (no-go trials only, orange) and execution (blue). Black triangles denote approximate location of sulci as anatomical landmarks.

As for potential differences between the subdivisions, we hypothesized that the relative size of the distances during planning and execution activity may differ between subregions. However, when we computed the ratio between distances during planning and execution, we did not find any major differences in the different subdivisions of M1 and S1 (see black line at the bottom of (Author response image 1) ; BA 4a: 0.23, BA 4b: 0.16, BA 3a: 0.24, BA 3b: 0.19, BA 1: 0.22, BA 2: 0.31; mean across subdivisions: 0.234 ± 0.019). Thus, this shows that the evidence for planning activity (in relationship to what is seen in execution) is roughly stable across subregions of S1 and M1. We included this analysis in the manuscript (page 7-8, line 143-205).

3) Role of ipsilateral hemisphereI would also suggest conducting the same type of analyses on S1 and M1 of the ipsilateral hemisphere as a meaningful control site. Moreover, this analysis could provide novel insights on the possible role of ipsilateral S1 in representing ipsilateral movements. Indeed, there is a growing body of evidence showing the representation of ipsilateral movements in the motor cortex (Bundy and Leuthardt, 2019), but much less is known on the role of S1. If a representation of the planned movements is present in one or more subsectors of S1 (see Gale et al. 2021), the comparison between the informative patterns across the two phases of the task (and maybe even across hemispheres) might be also of interest.

We agree that the role of ipsilateral S1 and M1 during planning and execution is a highly interesting question, albeit slightly orthogonal to the scope of our study. Indeed, in Berlot et. al (2018b) we speculated whether widespread representation of finger movements in the ipsilateral hemisphere could be explained by planning-related processes. Our current data only provide mixed evidence for this idea. When we performed the profile analysis (analogous to Figure 2E-2F) in the ipsilateral hemisphere we found slight deactivation around the central sulcus (comparable to the contralateral hemisphere) and very small (non-significantly different from zero) distances during planning (Author response image 2) . From this analysis, it is hard to conclude whether this null result speaks against a role of the ipsilateral hemisphere in planning or is simply due to lack of power / noise in the data.

**Author response image 2. sa2fig2:** A. Profile ROIs analysis of activation (mean percent signal change) in the ipsilateral (right) hemisphere for planning (orange) and execution (blue). Vertical dotted lines denote boundaries of ROI subdivisions. Black triangles denote the three main sulci in the right hemisphere: postcentral sulcus (post-CS), central sulcus (CS), and precentral sulcus (pre-CS). Horizontal bars indicate significance (p < 0.05) in a 2-sided one-sample t-test vs zero. B. Same as A but for multivariate crossnobis distance.

4) Whole brain univariate analyses and searchlight analysesI think it would be useful to provide whole brain data of the univariate and multivariate analyses for the two phases of the task. The authors might consider adding the results of these analyses as supplementary figures.Whole brain univariate results could provide an overview of the network of regions engaged during the planning of finger movements. Looking at Figure 1, it seems that there is only an anterior parietal region recruited by the task (possibly at an uncorrected level). I would expect to see activation in parieto-frontal regions. This analysis might also partially answer to my first point.I also would suggest conducting searchlight analyses for the two phases of the task. These analyses might serve different purpose. First, these results could provide insights on additional regions (e.g. parietal regions, SII) representing different planned finger movements. Second, as most (if not all) previous MVPA studies adopting paradigm with a delayed execution in the field investigated complex hand/arm actions, the present data might provide an indirect indication on which part of the parieto-frontal motor networks are engaged for simple finger movements and which are specific for more complex arm/hand actions.Finally, the comparison/conjunction between univariate and multivariate results might provide insights on the regions within the motor networks which are both engaged by the task and host a representation of the different movements. This analysis would be particularly interesting for the planning phase of the task.

We agree and now provide both univariate activation and multivariate searchlight whole brain (both hemispheres) maps for the two phases of the task (planning and execution) as supplementary figure (Figure 2 – supplement 2).

5) Comparison of the topography of informative patterns between the planning and execution phase of the taskI apologise in advance if this concern is related to my misinterpretation of this analysis, as I'm not familiar with the Hotelling T2 test (lines 130-132). My understanding of this test is that it allows to test the difference between two (multivariate) means of different distributions. I think that the authors compared the distribution of the crossnobis distance for M1 and S1 across subjects.If my understanding of the analysis is correct, then this analysis showed that the distribution of crossnobis distances within M1 and S1 is consistent across the two phases, but this analysis doesn't directly test if the spatial organization of the informative patterns across the 2 phases is different. I would suggest to directly compare the distribution of crossnobis distance between the two phases across subjects independently in each vertex of M1 and S1. The result of this analysis would be a spatial map of the difference between the distributions of the two phases. This would allow to appreciate if there is any spatial asymmetry between the 2 phases.As a general recommendation, I would suggest providing additional details in the description of this analysis to allow an easier understanding of its implications.

To improve the comparison of the topography of activity patterns between planning and execution, we additionally computed the ratio between planning and execution profiles (page 7-8, line 145-205). We also embraced the reviewer’s suggestion and expanded on the logic for this comparison (page 7, line 143-147).

Reviewer #3 (Recommendations for the authors):1. I urge the authors to include more details regarding the analysis they conducted. This is an overall recommendation, but a few things were specifically unclear to me:a. Since the PCM is not a commonly used analysis, the manuscript would especially benefit from more conceptual explanation regarding the rationale of conducting this analysis, further explanation on the method, and how the achieved r-values can be interpreted.

We expanded our methods to better explain the rationale for our analysis and the interpretation both in the result section (page 12-13, line 293-390) and in methods (page 23-24, line 717-758). We also have provided reference to the now extensive online documentation on the PCM toolbox and the fully-fledged example of the correlation analysis presented here.

b. What does the colourbar scaling represent in Figure 1B and C – is this percent signal change?

There is no colorbar in Figure 1B. If the reviewer is referring to Figure 2A-D, colorbars in A and C reflect percent signal change (as in 2E), whereas colorbars in B and D reflect distance (arbitrary units, as in 2F). We now clarified this information in the figure and figure caption.

c. Similarly – Are all coloured vertices in Figure 2A-D significantly activated/ have significant distances? Or what does a value of 0.1 and 0.001 represent? Also, are the maps in Figure 2A and C corrected for multiple comparisons? This is not reported in the Methods section or the figure legend.

All surface maps are intended for visualization purposes only and are not corrected for multiple comparisons. We based our conclusions on statistical tests performed at the ROI level (see Figure 2E-F, horizontal bars). The numbers below the colorbars are not *p*-values; rather, they indicate either percent signal change values (A-C) or crossnobis distance values (B-D).

d. What analysis was ran to test for informative patterns in M1 and S1 while correcting for the influence of behavioural patterns – this analysis is not reported in the methods section.

The analysis showing informative patterns in M1 and S1 while correcting for the influence of behavioural patterns can be seen in Figure 3C. Page 10, line 264-267: “More importantly, a significantly positive intercept in the linear fit in Figure 3C (M1: p = 0.032; S1: p = 0.007) shows that, even after correcting for the influence behavioral patterns, the activity patterns in M1 and S1 remained informative”. We now added the intercept p-values also in Figure 3C and this rationale in the methods section (page 23, line 712-715).

e. Did the authors correct for multiple comparisons across the ROIs tested?

We did not, but we limited the number of tests by only including two ROIs in our PCM analysis.

f. For the analysis presented Figs 2E and F, did the authors average across the y-coordinates on the whole brain flatmap, or were this values extracted from a single straight line? i.e. do the distances on the cortical surface represent averages or rather values from single vertices? This analysis is not mentioned in the Methods section.

The inset at the top of Figure 2 shows the inflated cortical surface of the contralateral (left) hemisphere and highlights the virtual strip used for the cross-section analysis (white). The outline of the white strip shows the area that was used to create the new profile ROIs that result in the values plotted on the y-axis of Figure 2E-F. We added this new analysis description to the methods section (page 22-23, line 684-696), and to the caption of Figure 2.

g. The PCM rationale is better explained in the Results section, from only reading the methods section the rationale behind this method was unclear.

We now added more information about the rationale for the PCM analysis to the methods section as well (page 23-24, line 718-732).

h. Regarding the PCM analysis: If a model of 0.4 correlation is most predictive, can we conclude the representational patterns are significantly correlated? Or how can we interpret these correlational models?

We can conclude that the patterns are significantly correlated because correlation models from 0.4 to 1 perform significantly better than the zero-correlation model. However, given that the likelihood of the data did not differ significantly among models with a correlation of > 0.4, we do not see any evidence to prefer one of these models over each other. This could be either because the true correlation is partial (less than 1), or simply due to measurement noise (even if the true correlation was in fact 1).

i. Regarding the PCM analysis: How could a best fitting model perform better from the one-correlation model (i.e. does it make sense to test achieved r>1)? Isn't a correlation of 1 the maximum achievable value? I think that if activity patterns during planning and execution are truly a mere scaled version of each other, the correlation between the activity patterns should be 1. So if the achieved r is sig less than 1, this would argue against the scaled version argument. This means that the authors should test if their model performs sig less than the one-correlation model.

The maximum correlation achievable is indeed 1. We did not test whether the best fitting model has a correlation > 1. Rather, we tested if the best fitting model (at the group level, found using a cross-validation, see Methods) had a higher log-likelihoods than the log-likelihoods for the 1-correlation model (Figure 4C). So, by “performs better”, we mean “has a higher likelihood”, not “has a higher correlation”. Our analysis shows that the best fitting model does not perform significantly better than the 1-correlation model. This is compatible with the view that planning and execution patterns are a scaled version of each other. We now clarify this in the text (page 13, line 380-384).

2. What could potentially be added to the argument against micromovements explaining neural distances in S1/M1 during movement planning is that the expected correlation would be positive, while you see a non-significant negative correlation. It would also be interesting to test whether there are significant differences between the behavioural and neural distances during planning and execution for each ROI. For me that would be more convincing than showing a mere non-significant correlation (without Bayesian stats).

We agree that the idea of micro-movements would have predicted a positive correlation. The argument that the correlation was non-significant, and even slightly negative, does not provide the strongest evidence against this hypothesis. Our conclusion that micromovements cannot explain the representations found is that, even if we control for the influence of micromovements, the distances both in M1 and S1 are significant. This is now clearly outlined in the text (page 10, line 264-267) and we added the significant p-value for the intercept to Figure 3C.